# PHALAR: Phasors for Learned Musical Audio Representations

**Davide Marincione** [1] **Michele Mancusi** [1 2] **Giorgio Strano** [1] **Luca Cerovaz** [1 3] **Donato Crisostomi** [1]
**Roberto Ribuoli** [1] **Emanuele Rodolà** [1 3]

## Abstract

Stem retrieval, the task of matching missing stems to a given audio submix, is a key challenge currently limited by models that discard temporal information. We introduce PHALAR, a contrastive framework achieving a relative accuracy increase of up to $\approx 70\%$ over the state-of-the-art while requiring $< 50\%$ of the parameters and a $7\times$ training speedup. By utilizing a Learned Spectral Pooling layer and a complex-valued head, PHALAR enforces pitch-equivariant and phase-equivariant biases. PHALAR establishes new retrieval state-of-the-art across MoisesDB, Slakh, and Choco-Chorales, correlating significantly higher with human coherence judgment than semantic baselines. Finally, zero-shot beat tracking and linear chord probing confirm that PHALAR captures robust musical structures beyond the retrieval task.

## 1. Introduction

Modern representation learning for audio has largely adopted paradigms from computer vision, treating spectrograms as static 2D images processed by standard CNNs or Vision Transformers. A cornerstone of these architectures is the use of pooling operations, such as Global Average Pooling (GAP), to enforce translational invariance. While invariance is desirable for semantic classification (e.g., identifying that a clip contains a "guitar" regardless of when it plays), it is detrimental for tasks requiring structural coherence, such as music mixing and stem separation.

In this work, we focus on the specific problem of modeling musical coherence: given a partial mix (e.g., drums and bass), the objective is to identify which missing stems temporally and harmonically fit with it. This differs cate-

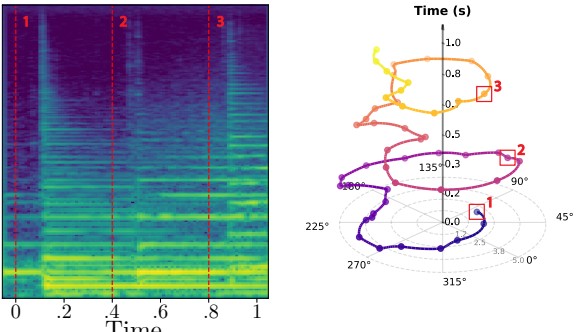

*Figure 1.* **Emergent Phase-Equivariance.** Our model's Learned Spectral Pooling layer maps temporal alignment to geometric rotation in the complex plane. *Left:* Three timesteps $(1, 2, 3)$ at identical offsets from note onsets. *Right:* Time-expanded polar plot of a learned feature. As time progresses, the feature revolves around the origin. Because the model is phase-equivariant, positions with the same relative timing (red boxes) share the same phase angle regardless of their absolute time. This allows PHALAR to resolve rhythmic coherence where standard magnitude-based models fail.

gorically from standard semantic tasks, where the goal is merely to recognize what is present. The challenge is that coherence is strictly dependent on temporal alignment: two signals can contain the exact same instruments yet be entirely incoherent if misaligned (e.g., drums slightly off-beat with the bass). Foundation models like CLAP (Wu* et al., 2023) and CDPAM (Manocha et al., 2021), designed for semantic similarity, are thus engineered to be "structurally blind": their reliance on GAP discards temporal ordering, collapsing distinct rhythmic alignments into identical latent representations. Even COCOLA (Ciranni et al., 2025), which targets harmonic compatibility, relies on GAP, limiting its ability to capture fine-grained rhythmic phase.

This paper proposes a fundamental shift from temporal/phase invariance to equivariance. We observe that while the magnitude spectra of musical signals are shift-equivariant in time, standard real-valued networks lack the structure to manipulate this shift explicitly. Leveraging the Fourier Shift Theorem, we recognize that a temporal translation in the input domain corresponds to a phase rotation in the frequency domain. Consequently, to explicitly model coherence, an aggregation scheme must preserve tempo-

[1]Department of Computer Science, Sapienza University of Rome, Italy [2]Moises Systems, Inc. [3]Paradigma, Inc.. Correspondence to: Davide Marincione <marincione@di.uniroma1.it>, Michele Mancusi <michele.mancusi@moises.ai>.

ral alignment by construction. We achieve this by shifting the representation space from real-valued magnitudes to complex-valued phasors; see Figure 1.

To this end, we introduce PHALAR (Phasors for Learned Musical Audio Representations), a contrastive learning framework tailored for musical coherence. PHALAR decouples feature extraction from alignment by employing a real-valued axial backbone to extract harmonic features, followed by a Learned Spectral Pooling layer that projects these features into the complex frequency domain. This allows temporal positions to be encoded as phase angles, which are then preserved by a phase-equivariant Complex-Valued Neural Network (CVNN) projection head.

Our contributions can be summarized as follows:

- We propose PHALAR, a novel contrastive audio framework that explicitly decouples harmonic content from rhythmic alignment.

- We set a new state-of-the-art in stem-to-mix retrieval, achieving a relative increase of up to $\approx 70\%$ in accuracy over the previous state-of-the-art (Ciranni et al., 2025), while requiring $< 50\%$ of the parameters and offering a $7\times$ training speedup (50 vs. 340 GPU-hours).

- We demonstrate a fundamental orthogonality between similarity and coherence modeling. While similarity-based foundation models (Wu* et al., 2023; Manocha et al., 2021) perform at random chance on coherence tasks, PHALAR correlates significantly with human perception.

We release our code, checkpoints and human evaluation results at github.com/gladia-research-group/phalar.

## 2. Related Works

### 2.1. Contrastive Representation Learning in Audio

Self-supervised learning has become the standard for audio representation, primarily via contrastive objectives to maximize agreement between augmented views of inputs. Early approaches in speech, such as Wav2Vec 2.0 (Baevski et al., 2020) and HuBERT (Hsu et al., 2021), demonstrated the efficacy of masked prediction. In the music domain, CLMR (Spijkervet & Burgoyne, 2021) and MERT (Li et al., 2024) adapted SimCLR-style (Chen et al., 2020) frameworks, utilizing augmentations like pitch shifting and EQ to enforce invariance to recording conditions. More recently, large-scale foundation models like CLAP (Wu* et al., 2023) and AudioLDM (Liu et al., 2023) have leveraged joint audio-text embedding spaces trained on large scale datasets.

A pervasive limitation in these architectures is their aggregation mechanism. To produce fixed-size embeddings from variable-length inputs, models predominantly rely on Global Average Pooling (GAP) or classification tokens (Devlin et al., 2019; Dosovitskiy et al., 2021). While effective for semantic classification, these operations *enforce* translation invariance, marginalizing the temporal structure and phase information critical for time-sensitive tasks.

### 2.2. From Semantic Similarity to Structural Coherence

Existing audio evaluation metrics are designed to assess semantic similarity or generation quality, rather than structural coherence. Distribution-based metrics like Fréchet Audio Distance (FAD) (Kilgour et al., 2019) rely on embeddings from semantic classifiers (Wu* et al., 2023; Hershey et al., 2017; Li et al., 2024; Kumar et al., 2023; Défossez et al.) to measure domain approximation, while sample-level metrics like ViSQOL (Chinen et al., 2020) quantify spectral similarity to a reference. Neither paradigm explicitly captures the temporal interplay between sources.

Historically, harmonic and rhythmic alignment have been the focus of specialized IR tasks like beat tracking (Cheng & Goto, 2023). In deep representation learning, COCOLA (Ciranni et al., 2025) recently attempted to score harmonic compatibility, yet its reliance on real-valued global pooling limits its ability to capture fine-grained rhythmic phase. Other reference-free metrics, like Audiobox-Aesthetics (Tjandra et al., 2025), provide absolute "likability" scores. While useful for filtering data, these scores are agnostic to the relative alignment of multiple sources and thus fail as coherence measures.

### 2.3. Complex-Valued Neural Networks (CVNNs)

CVNNs extend deep learning to the complex domain, respecting the algebra of phasors and wave physics. Trabelsi et al. (2018) formalized the necessary building blocks, including complex convolutions and initializations. These architectures have achieved state-of-the-art results in speech enhancement and source separation (Choi et al., 2018), where explicit phase reconstruction is critical.

However, to date, the application of CVNNs has been largely restricted to generative tasks (reconstruction/denoising) (Cerovaz et al., 2026). Their utility in discriminative representation learning remains under-explored in audio. In other domains, complex-valued embeddings have shown promise; for instance, knowledge graph methods like RotatE (Sun et al., 2019; Trouillon et al., 2016) model relations as rotations in the complex plane to capture anti-symmetric and inversion patterns, and other works (Li et al., 2018) have shown NLP applications.

We posit that music, being fundamentally periodic, is the ideal modality for such geometric biases. We bridge this gap by applying complex-valued metric learning to capture

temporal shifts as phase rotations.

# 3. Method

Rather than processing raw complex spectrograms directly (Cerovaz et al., 2026), PHALAR first extracts harmonic features from magnitude spectra via a real-valued backbone. Then, it achieves temporal sensitivity through a **Learned Spectral Pooling** layer: it applies a Fourier transform across the temporal dimension of the extracted feature maps. By the Shift Theorem, this operation maps the relative timing of features to phase rotations in the complex domain. A CVNN head then processes these latents to assess alignment.

This architecture enforces two specific inductive biases:

- **Pitch-Equivariance & Awareness:** The backbone extracts interval-aware features via pitch-equivariant convolutions on CQT inputs, which are subsequently mapped to absolute pitch-aware embeddings during spectral pooling.

- **Phase-Equivariance:** Established by the spectral pooling layer, which converts temporal shifts into phase information for the CVNN to evaluate.

## 3.1. Harmonic Backbone

The backbone is a lightweight 2D CNN optimized for harmonic feature extraction and computational efficiency.

PHALAR processes Constant-Q Transform (CQT) (Holighaus et al., 2012) spectrograms; unlike Mel-spectrograms, the CQT's logarithmic spacing ensures that pitch shifts are purely linear translations. This allows our kernels to recognize harmonic intervals (e.g., a "major third") identically across all keys, a powerful inductive bias that eliminates the need to learn key-specific variations in the backbone.

The architecture has 10 layers, each with an axial residual design to decouple spectral and temporal processing:

1. Frequency-wise Convolutions ($3 \times 1$): Isolate and extract harmonic relationships within individual timesteps.

2. Time-wise Convolutions ($1 \times 3$): Capture the temporal evolution of frequency bins.

3. Point-wise Convolutions ($1 \times 1$): Facilitate feature mixing and channel-wise projection.

To manage computational overhead, every even layer employs a strided time-wise convolution, resulting in a total temporal compression factor of $32\times$ before the data reaches the spectral pooling stage.

## 3.2. Spectral Aggregation

To preserve critical timing, we replace standard GAP with **Learned Spectral Pooling**. This operation maps temporal sequences into the frequency domain, adapting the down-sampling technique from Rippel et al. (2015) to ensure synchronization cues are not discarded.

Unlike the translational invariance of GAP (Manocha et al., 2021; Saeed et al., 2020; Ciranni et al., 2025), which effectively marginalizes the temporal structure, our spectral approach transforms temporal relationships into phase rotations for the model head to process.

### 3.2.1. TEMPORAL TO SPECTRAL PROJECTION

Let $\mathbf{X} \in \mathbb{R}^{B \times H \times F \times T'}$ denote the feature map from the backbone, where $H$ is the channel depth and $T' = \lceil T/32 \rceil$ is the compressed time dimension. We flatten the channel and frequency dimensions to obtain a unified feature space $\bar{\mathbf{X}} \in \mathbb{R}^{B \times (HF) \times T'}$. To extract semantic features prior to pooling, we project $\bar{\mathbf{X}}$ onto a learned basis $\mathbf{W}_{\text{proj}} \in \mathbb{R}^{(HF) \times D}$. This projection operates pointwise in time,

$$\mathbf{Z}_{\text{time}} = \bar{\mathbf{X}}\mathbf{W}_{\text{proj}} \in \mathbb{R}^{B \times T' \times D} . \tag{1}$$

Because this projection operates simultaneously over all frequency bins $F$, $\mathbf{Z}_{\text{time}}$ encodes both the harmonic interval structure (from the backbone) and the absolute frequency position of those intervals. This two-stage design (equivariant extractor followed by a full-frequency projection) ensures the model becomes explicitly pitch-aware at the point of spectral pooling.

We then apply a Real Fast Fourier Transform (RFFT) (Brigham & Morrow, 1967) along the temporal axis to obtain the spectral representation

$$\mathbf{S} = \text{rfft}(\mathbf{Z}_{\text{time}}) \in \mathbb{C}^{B \times C \times D} , \tag{2}$$

where $C = \lfloor T'/2 \rfloor + 1$. By truncating or padding to a fixed $C$, we obtain a fixed-size embedding.

In our implementation, the projection matrix $\mathbf{W}_{\text{proj}}$ maps the flattened backbone features to $D = 80$ dimensions, and we fix the temporal frequency cutoff at $C = 8$. Consequently, each embedding contains exactly $D \times C = 640$ complex values, yielding a latent footprint equivalent to 1280 real values; deliberately chosen to match the bottleneck dimensionality of our primary baseline, COCOLA(Ciranni et al., 2025), to ensure a fair comparison of architectural efficiency.

In this representation, the *magnitude* $|\mathbf{S}_{c,d}|$ encodes the prevalence of a specific harmonic pattern (e.g., a "snare hit shape") $d$ at a modulation frequency $c$, while the *phase* $\angle\mathbf{S}_{c,d}$ explicitly encodes its temporal shift.

This operation can be interpreted as a learnable variant of the Modulation Spectrum (Atlas & Shamma, 2003). Unlike classical modulation analysis which operates on raw

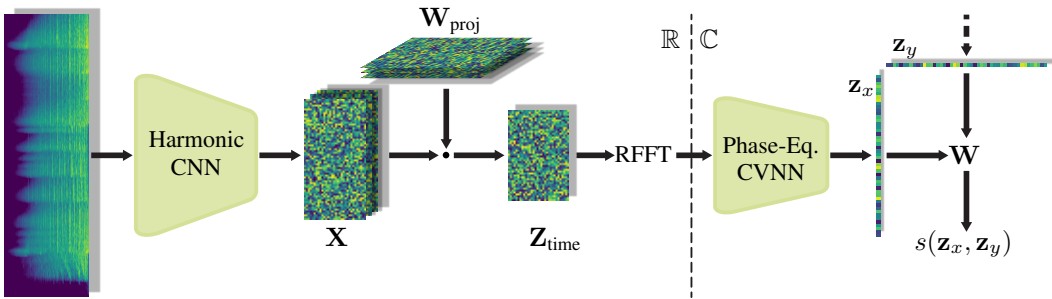

*Figure 2.* Depiction of PHALAR's architecture: a spectrogram is fed to the CNN, the resulting feature map is projected onto a learned basis and processed via Fast-Fourier Transform. The complex-valued result is then refined by the phase-equivariant CVNN, and, at the end, a score is computed between two sample embeddings.

spectrogram frequencies, PHALAR computes modulation over learned semantic features. This allows the model to disentangle the rhythmic profile of specific instruments (e.g., the groove of a bassline) from the global mix, converting the temporal alignment problem into a geometric relationship in the complex plane.

### 3.3. Complex-Valued Projection Head

Since $\mathbf{S}$ is complex-valued, standard real-valued MLPs cannot process it without destroying the phase structure (and thus the alignment information). We implement a CVNN (Trabelsi et al., 2018) projection head where every operation is *phase-equivariant*, satisfying $f(x \cdot e^{i\theta}) = f(x) \cdot e^{i\theta}$.

Specifically, the head consists of a sequence of two complex linear layers. To allow the model to learn non-linear feature interactions while strictly preserving temporal alignment, the first linear layer is followed by a Complex RMSNorm and a phase-preserving modReLU. The mathematical formulations for these components are detailed in Section A. This CVNN head projects the 640-dimensional complex input down to a final output dimension of 512 complex values.

#### 3.3.1. PHASE-AWARE BILINEAR SIMILARITY

To quantify structural coherence, we employ a similarity metric tailored to the algebraic properties of our spectral phasors. Specifically, we define the score as the real part of a parametrized Hermitian inner product between $L_2$-normalized feature vectors $\mathbf{z}_x, \mathbf{z}_y \in \mathbb{C}^D$:

$$s(\mathbf{z}_x, \mathbf{z}_y) = \Re(\mathbf{z}_x^H \mathbf{W} \mathbf{z}_y), \tag{3}$$

where $\mathbf{W} \in \mathbb{C}^{D \times D}$ is a learnable complex weight matrix. By taking the real part, we project the complex-valued alignment into a scalar score suitable for contrastive objectives while ensuring the model remains sensitive to the relative phase shifts encoded within the embeddings.

This formulation offers a distinct advantage over real-valued dot-products, as the complex weights allow the model to

apply *learnable phase rotations*. This mechanism enables the model to "align" stems by rotating their phase to account for consistent micro-timing deviations, such as a "laid back" groove, thereby maximizing the coherence score.

Furthermore, we intentionally omit saturating non-linearities like tanh found in related works (Saeed et al., 2020; Ciranni et al., 2025). With a linear output, we ensure that high-energy transients contribute proportionally more to the final score than low-energy background noise.

**Symmetric Inference**  The bilinear form in Equation (3) is non-commutative and, while asymmetric scoring is permissible during contrastive training, retrieval tasks require a symmetric metric. During inference, we therefore use:

$$s_{\text{comm}}(\mathbf{z}_x, \mathbf{z}_y) = \frac{s(\mathbf{z}_x, \mathbf{z}_y) + s(\mathbf{z}_y, \mathbf{z}_x)}{2}. \tag{4}$$

## 4. Experiments

We evaluate PHALAR on the task of **Stem Retrieval**: given a query submix (e.g., drums + bass), the model must identify the complementary submix (e.g., vocals + guitar) from the same original track among a set of distractors. This task acts as a proxy for structural coherence, requiring the model to resolve precise rhythmic and harmonic alignments rather than semantic categories.

Our experiments demonstrate the following:

- PHALAR achieves a relative increase of up to $\approx 70\%$ in retrieval accuracy over current benchmarks while utilizing less than half the parameters (Section 4.2);

- Our axial backbone and specialized pooling facilitate a $7\times$ training speedup compared to previous coherence-oriented models (Section 4.1);

- Phase-aware embeddings provide the highest correlation with human coherence judgment, identifying

structural failures that "coherence-blind" foundation models miss (Section 4.3);

- Despite no explicit supervision for rhythm or pitch, PHALAR's inductive biases enable zero-shot beat tracking and linear chord probing (Section 4.6).

### 4.1. Experimental Setup

**Datasets & Sampling** We construct a composite dataset integrating `MoisesDB` (Pereira et al., 2023) (using a random $0.8/0.1/0.1$ split at track level), `Slakh2100` (Manilow et al., 2019), and `ChocoChorales` (Wu et al., 2022). To enforce structural coherence, we generate training pairs dynamically: for a given music track, we generate two time-aligned *disjoint submixes* $\mathbf{x}_A$ and $\mathbf{x}_B$ such that the set of instruments in $\mathbf{x}_A$ is mutually exclusive to those in $\mathbf{x}_B$ (e.g., if "Vocals" are in the anchor, they cannot be in the positive). This prevents the model from relying on trivial identity mapping of specific instrument timbres.

**Optimization & Efficiency** Models are trained for 80k steps with a batch size of 64 on two NVIDIA A100 GPUs using the `Muon` optimizer (Jordan et al., 2024), with learning rates $\eta_{\text{muon}} = 0.02$ and $\eta_{\text{adam}} = 4 \times 10^{-3}$. To isolate architectural gains from optimization benefits, we upgraded and retrained the COCOLA baseline using Muon for an equivalent duration. PHALAR demonstrates superior efficiency, completing training in 50 GPU-hours compared to COCOLA's 340 GPU-hours. This $7\times$ **speedup** is driven by our parameter-efficient axial backbone and the elimination of CPU-bound Harmonic-Percussive Separation (Fitzgerald, 2010; Driedger et al., 2014) pre-processing.

**Label Smoothing for Sampling Collisions** Standard InfoNCE (Oord et al., 2018) training assumes all negatives in a batch are true negatives, an assumption frequently violated in music where different tracks may share the same key, tempo, or genre. Penalizing these pairs introduces gradient noise. To address this, we apply Label Smoothing (Szegedy et al., 2016), relaxing the positive pair's target probability to $l = 0.9$. Distributing the residual mass among negatives prevents the model from over-separating tracks that are harmonically compatible despite being distinct.

**Augmentation** To ensure robustness to recording conditions, we apply the on-the-fly augmentations: random crop $T \in [2, 10]$s (applied identically to both submixes to preserve their beat alignment), gain stage $\pm 6$ dB, and additive noise injection (white, pink, brown, and transient bursts).

**Baselines** We compare PHALAR against:

- **COCOLA:** (Ciranni et al., 2025) The current state-of-the-art for coherence; a real-valued CNN with GAP.

*Table 1.* **Contrastive retrieval** ($\uparrow$) We report Top-1 accuracy on disjoint submix retrieval. ($\dagger$ =fine-tune with Learned Spectral Pooling and CVNN head)

| Dataset | K | PHALAR (2.3M) | COCOLA (5.2M) | MERT† (95M) | CLAP (200M) | CDPAM (26.2M) |
|---|---|---|---|---|---|---|
| MoisesDB | 8 | **86.79** | 75.81 | 67.39 | 12.85 | 11.15 |
| MoisesDB | 16 | **81.49** | 64.44 | 59.13 | 6.19 | 5.03 |
| MoisesDB | 64 | **70.87** | 41.84 | 45.85 | 1.24 | 1.15 |
| Slakh2100 | 8 | **87.69** | 79.33 | 66.70 | 10.91 | 11.45 |
| Slakh2100 | 16 | **83.28** | 71.58 | 58.39 | 5.12 | 5.83 |
| Slakh2100 | 64 | **72.37** | 55.84 | 46.13 | 1.62 | 1.76 |
| ChocoChorales | 8 | **99.65** | 97.82 | 96.49 | 10.72 | 7.54 |
| ChocoChorales | 16 | **99.45** | 96.02 | 93.79 | 4.09 | 3.02 |
| ChocoChorales | 64 | **98.61** | 89.34 | 86.65 | 0.71 | 0.59 |

- **MERT:** A state-of-the-art music understanding foundation model. To provide the strongest possible foundation baseline: we extract frozen MERT embeddings and process them using our novel Learned Spectral Pooling and CVNN head.

- **CLAP:**[1] (Wu* et al., 2023) A foundation model trained for text-audio retrieval, representing state-of-the-art semantic embedding. We include it not as a direct competitor, but as a diagnostic probe to test the structural awareness of semantic representations.

- **CDPAM:** (Manocha et al., 2021) A deep perceptual audio similarity metric, similarly included as a probe to contrast perceptual similarity with structural coherence.

- **ViSQOL:** (Chinen et al., 2020) A standard metric for reference-based audio quality estimation.

- **Audiobox-Aesthetics:** (Tjandra et al., 2025) A deep, reference-free, audio quality metric that provides absolute scores for quality.

### 4.2. SOTA in Contrastive Retrieval

We measure performance using $K$-way Contrastive Retrieval Accuracy. As shown in Table 1, PHALAR establishes a new state-of-the-art across all datasets. The architectural advantage of phase-equivariance is most evident at $K = 64$, where the task becomes significantly harder due to the increased probability of tonal collisions (distractors with similar keys). On MoisesDB, PHALAR achieves a relative **improvement of** $+\mathbf{69}\%$ over the COCOLA baseline ($71\%$ vs $42\%$), with half its parameters (2.3M vs 5.2M).

**Orthogonality of Coherence and Similarity** A key finding of our study is the disconnect between perceptual/semantic similarity and structural coherence. Foundation models like CLAP are trained to map audio to text descriptions (e.g., "a rock song"), enforcing invariance to

---

[1]Specifically, `music_audioset_epoch_15_esc_90.14.pt`.

specific tempos or key signatures. When utilized as diagnostic probes on the stem retrieval task, CLAP and CD-PAM effectively collapse to random chance (e.g., $\approx 1.2\%$ at $K = 64$). To investigate whether this is strictly an aggregation issue, our MERT baseline equips a 95M-parameter foundation model with our phase-aware spectral pooling head. While this geometric bias allows MERT to successfully extract coherence information and surpass COCOLA (reaching 45.85 on MoisesDB $K = 64$), it still falls $\approx 25$ points short of PHALAR. This demonstrates two things: first, modeling the *interactions* between sources requires a fundamentally different geometric inductive bias than semantic classification; second, achieving true state-of-the-art structural coherence requires the end-to-end alignment of a pitch-equivariant backbone and a complex-valued head, rather than retrofitting massive foundation models. Full ablation studies on MERT aggregation strategies are provided in Section C.

### 4.3. Human-Centric Validation

While contrastive retrieval accuracy measures the ability to identify the *exact* ground truth, it does not strictly quantify perceptual quality. A robust audio representation should define a metric space where distance correlates with perceptual coherence: a "bad" submix should be far from the mix, and a "good" submix (even if generated) should be close.

To validate this, we conducted a subjective listening test correlating human coherence ratings with the embedding distances computed by PHALAR and baselines.

**Listening Test Protocol** We curated a dataset of 98 audio samples (49 Bass, 49 Drums) from the `MUSDB18-HQ` (Rafii et al., 2017; 2019) test set. For each sample, we generated three variations of the missing stem using stem-generation models of varying quality: Moises' stem generator (commercial SOTA), STAGE (Strano et al., 2025), and StableAudio-ControlNet (Evans et al., 2025). Including the Ground Truth, this yielded 4 variations per track, creating a diverse spectrum of coherence ranging from artifacts/misaligned generations to studio-quality mixes.

We recruited $N = 22$ participants, each blindly evaluating 10 random cases. For every case, participants rated 4 variations on a Likert scale of 1 (Incoherent/Clashing) to 5 (Perfectly Coherent). This resulted in 880 individual ratings.

**Correlation with Human Perception** We computed the correlation between standardized human ratings (z-scored per user to normalize subjective baselines) and the similarity scores produced by the models.

As detailed in Table 2 and Figure 3, PHALAR achieves the highest alignment with human judgment across both Pearson ($\rho$) and Spearman ($r_s$) coefficients. To rigorously

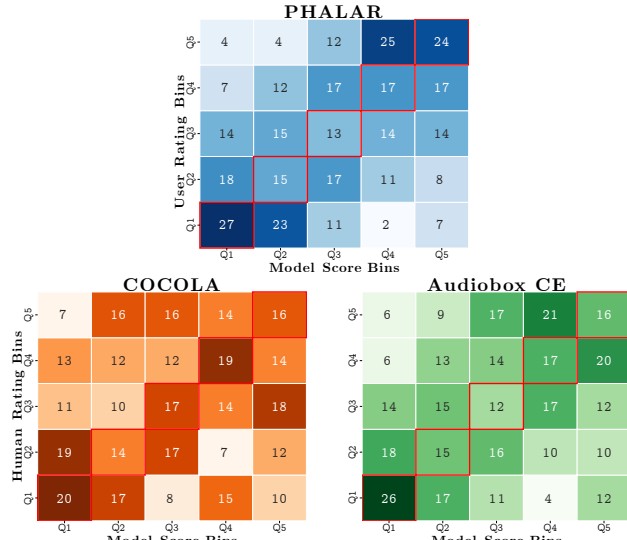

*Figure 3.* **Human v. Model score** Heatmaps over PHALAR, COCOLA and Audiobox CE's ratings' quintiles against averaged user opinions' quintiles.

*Table 2.* **Human-Model Comparison Stats** Steiger's test indicates significance between PHALAR and the respective baseline. For AIC lower is better.

| Model | Pearson $\rho$ ($\uparrow$) | Spearman $r_s$ ($\uparrow$) | Steiger $p$-val vs. PHALAR | AIC ($\downarrow$) |
|---|---|---|---|---|
| CLAP | 0.111 | 0.122 | $\leq 0.001$ | 2528.46 |
| CDPAM | $-0.015$ | $-0.011$ | $\leq 0.001$ | 2543.79 |
| ViSQOL | $-0.091$ | $-0.069$ | $\leq 0.001$ | 2538.13 |
| COCOLA | 0.181 | 0.153 | $\leq 0.001$ | 2519.36 |
| Audiobox$_{PC}$ | $-0.129$ | $-0.120$ | $\leq 0.001$ | 2540.00 |
| Audiobox$_{PQ}$ | 0.253 | 0.254 | 0.041 | 2501.53 |
| Audiobox$_{CU}$ | 0.236 | 0.247 | 0.022 | 2493.81 |
| Audiobox$_{CE}$ | 0.289 | 0.284 | 0.123 | 2476.89 |
| **PHALAR** | **0.387** | **0.414** | - | **2451.48** |

test these improvements, we employed Steiger's Z-test for dependent correlations. The results confirm that PHALAR's correlation is significantly higher than all baselines ($p < 0.05$), with the exception of Audiobox$_{CE}$ ($p = 0.123$), the score that predicts Content Enjoyment.

**Linear Mixed Effects Analysis** To account for subject-specific variability (e.g., some users generally rating higher than others), we modeled the data using a Linear Mixed Model (LMM)

$$R_{ij} = \beta_0 + \beta_1 S_{ij} + \beta_2 T_j + u_i + \epsilon_{ij} \quad u_i \sim \mathcal{N}(0, \sigma_u^2) \quad (5)$$

where $R_{ij}$ is the rating by user $i$ on item $j$, $S_{ij}$ is the model's score, $T_j$ the item's type ("bass" or "drums" categories), and $u_i$ is the random intercept per user.

We compare models using the Akaike Information Criterion (AIC), which estimates the relative quality of statistical models for a given dataset (although AIC accounts for model complexity, in this case it is irrelevant, as all LMMs are

*Table 3.* **System-Level Evaluation.** Aggregated PHALAR scores compared to Human Ratings and Fréchet Audio Distance (FAD).

| Model | Users (↑) | PHALAR (↑) | $\text{FAD}_{\text{CLAP}}$ (↓) | $\text{FAD}_{\text{MERT}_7}$ (↓) |
|---|---|---|---|---|
| Ground Truth | 3.86 | 5.66 | - | - |
| Moises | 3.04 | 5.53 | 0.350 | 10.6 |
| STAGE | 2.77 | 3.12 | 0.427 | 12.5 |
| SA-ControlNet | 2.55 | 3.01 | 0.564 | 10.7 |

the same, just with different fitting data). As shown in Table 2, PHALAR achieves the significantly lowest AIC. This confirms that, even when controlling for user variance, PHALAR provides the most explanatory power for predicting human perception of musical coherence.

**Comparison with Set-Level Metrics**   Fréchet Audio Distance (FAD) (Kilgour et al., 2019; Gui et al., 2024) is the industry standard for evaluating generative audio. However, we argue it is ill-suited for assessing coherence due to two fundamental limitations:

1. **Marginal vs. Conditional:** FAD measures the distance between the *marginal* distribution of generated and ground-truth stems. It assesses whether a sample sounds realistic, but ignores the *conditional* requirement: does it fit the specific backing track?

2. **Granularity:** FAD is a set-level metric requiring large sample sizes, rendering it useless for scoring individual inference results.

Table 3 highlights this limitation by comparing the rankings of three generative models: Moises, STAGE, SA-ControlNet. While standard $\text{FAD}_{\text{MERT}_7}$ fails to align with human judgment, incorrectly ranking SA-ControlNet (10.7) above STAGE (12.5), the aggregated PHALAR score reproduces the exact human ranking order (2.77 for STAGE vs. 2.55 for SA-ControlNet). By acting as a reference-aware metric that evaluates generated stems against their specific complementary mixtures, PHALAR captures the structural failures, such as rhythmic drift, that distribution-based metrics routinely miss.

### 4.4. Ablation Study

To rigorously disentangle the contributions of our architectural components and inductive biases, we perform a leave-one-out ablation study. We isolate four critical design choices: the harmonic input representation, the pooling mechanism, the phase-aware processing, and the metric space. The results are summarized in Table 4.

**The Necessity of Phase Equivariance**   Standard audio models typically rely on magnitude features, discarding phase information. To test this, we replaced our complex-valued head with a real-valued MLP operating solely on

*Table 4.* Leave-one-out ablation study over the PHALAR architecture. Results relative to MoisesDB $K = 64$ test.

| Model Variant | Accuracy (↑) | Drop |
|---|---|---|
| **PHALAR (Full)** | **70.87** | - |
| *w/o Spectral Pooling* | | |
| (Global Avg Pool + Real MLP) | 51.97 | $-18.9\%$ |
| *w/o Phase Equivariance* | | |
| (Magnitude Only + Real MLP) | 60.59 | $-10.3\%$ |
| (Complex Cosine Similarity) | 61.93 | $-8.94\%$ |
| *w/o Indefinite* $\mathbf{W}$ *during training* | | |
| (Positive Semi-Definite $\mathbf{W} = \mathbf{LL}^H$) | 67.85 | $-3.02\%$ |
| (Hermitian $\mathbf{W} = \mathbf{L} + \mathbf{L}^H$) | 69.92 | $-0.95\%$ |
| *w/o Strict Pitch Equivariance* | | |
| (Mel-Spectrogram Input) | 69.21 | $-1.66\%$ |

spectral magnitude. As shown in Table 4, this caused a catastrophic performance drop of $\mathbf{10.3}\%$. This confirms that magnitude alone cannot resolve rhythmic alignment; rather, the relative phase angles preserved by PHALAR are essential for detecting musical coherence.

We then evaluated the Complex Cosine Similarity, defined as the magnitude of the Hermitian inner product $|\mathbf{z}_x^H \mathbf{z}_y|$. While this metric operates in the complex domain, its mathematical invariance to global phase rotation results in poor performance. This validates that the model must strictly enforce phase alignment via Equation (3), rather than simply matching feature content up to an arbitrary rotation.

**Geometry of the Metric Space**   We investigated the algebraic properties of the learned weight matrix $\mathbf{W}$ compared to a strict Hermitian Positive Semi-Definite (PSD) formulation ($\mathbf{W} = \mathbf{LL}^H$). The PSD formulation degraded performance by $\approx 3\%$, suggesting the latent space benefits from an indefinite metric structure. While test-time averaging symmetrizes the matrix ($\mathbf{W}_{\text{eff}} = \frac{1}{2}(\mathbf{W} + \mathbf{W}^H)$), it does not enforce positive semi-definiteness. This flexibility allows the model to capture destructive interference; unlike a PSD matrix, which acts as a non-negative energy measure, an indefinite matrix can assign negative similarity scores to anti-aligned phase relationships.

At the same time, we also trained the model such that to parametrize $\mathbf{W}$ as Hermitian (and thus inducing commutativity in Equation (3) without the need for Equation (4)), but we found it to not increase accuracy in the model.

**CQT vs. Mel-Spectrograms**   Replacing the CQT with standard Mel-spectrograms decreases accuracy by $1.66\%$. While Mel-scales offer approximate shift-equivariance, they lack the geometric rigidity of the CQT. In a Mel-spectrogram, the spectral "shape" of a harmonic relation varies slightly across octaves due to filter-bank overlaps and resolution differences. The CQT's strict log-spacing acts as a stronger inductive bias, allowing the model to decouple

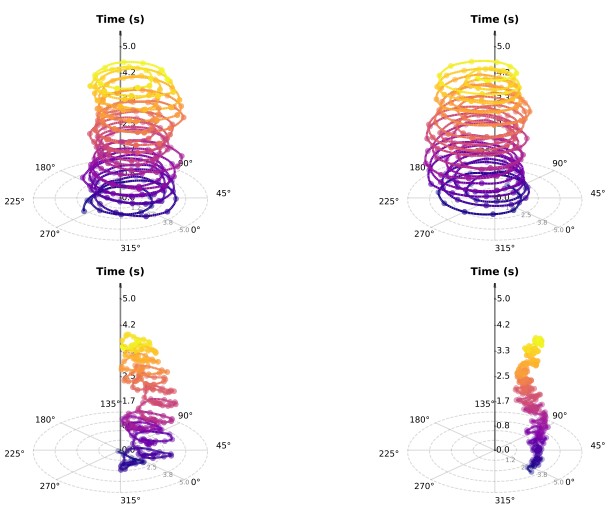

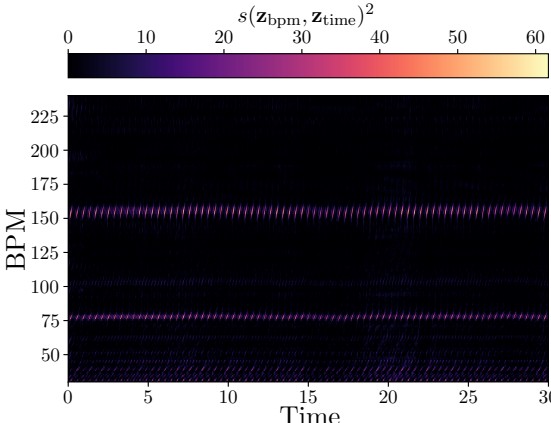

*Figure 5.* **Synthetized metronome BPMs v. Song embeddings** Heatmap of squared similarities between embeddings of a synthetic metronome at different BPMs and embeddings from the first 30s of "I Want to Live" (Slavov, 2023). Strong horizontal bands at 77 BPM and its first harmonic (154 BPM) precisely recover the ground-truth tempo, confirming that PHALAR linearizes rhythmic periodicity into detectable interference patterns *without temporal supervision*.

*Table 5.* **Beat tracking on GTZAN** Statistics computed via the `mir_eval` script at distance threshold of 70ms. PHALAR's phase-equivariance allows it to recover the track tempo ($F1 = 0.627$) as a geometric primitive, *despite never being supervised for rhythm*.

| Model | Precision (↑) | Recall (↑) | F1 (↑) |
|---|---|---|---|
| PHALAR | 0.717 | 0.587 | 0.627 |
| Beat This! | 0.893 | 0.905 | 0.888 |

*Figure 4.* Time-expanded polar plots reveal how the model partitions information. *Top:* "Rotating" features revolve about the origin, capturing periodic rhythmic structures through continuous phase cycles. *Bottom:* "Magnitude-Only" features are noncentered and oscillate within a restricted phase range. These emerge to represent global, time-agnostic attributes, such as key or mood, where precise temporal alignment is not required.

"harmonic interval" from "absolute pitch" more effectively.

### 4.5. Analyzing the Learned Pooling Layer

Figure 1 illustrates the phase-aware behavior of PHALAR, showing a specific feature maintaining a consistent phase value ≈ 100ms before string plucks. Combined with Section F, this confirms the model effectively exploits time-aware information as designed. Further analysis (in Figure 4) identifies two types of archetypal features:

- "Rotating" features, which complete revolutions about the origin-axis.

- "Magnitude-Only" features, which oscillate within a limited phase range and do not revolve about the origin.

We hypothesize that these "Magnitude-Only" features emerge to represent time-agnostic qualities like mood and key, where precise temporal alignment is unnecessary. While this provides an intuitive geometric interpretation of the latent space, it is a conjecture; rigorous confirmation would require correlating the phase variance of individual feature dimensions with key- and mood-labeled datasets.

### 4.6. Emergent rhythmic and harmonic structures

To empirically validate that PHALAR preserves musical structure without explicit supervision, we design two experiments targeting Rhythm (Phase) and Harmony (Magnitude).

#### 4.6.1. ZERO-SHOT BEAT TRACKING

We design a probing experiment using Zero-Shot Beat Tracking, validating that PHALAR maintains temporal alignment (rather than just texture).

**Method** We synthesize "probe" metronome tracks at various BPMs (30 − 240) and compute their similarity with the target track's embeddings. As shown in Figure 5, when the probe BPM matches the track's tempo, distinct interference patterns (vertical "stripes") emerge in the similarity matrix. By extracting the envelope of these correlations and passing them to a standard peak-picking algorithm (`librosa.beat_track`), we can recover the beat.

**Results** Table 5 compares this heuristic against a fully supervised SOTA baseline, Beat This! (Foscarin et al., 2024). While the supervised model naturally yields higher precision, PHALAR achieves a respectable F1-score of 0.627 **without ever seeing a beat label**. This confirms PHALAR successfully linearizes temporal relations, converting "alignment" into a geometric primitive (phase rotation).

*Table 6.* **Chord Linear Probing results** Results calculated at 95% confidence intervals across 5-fold cross-validation runs.

| Model | Accuracy ($\uparrow$) |
|---|---|
| Random | $1/25 = 4\%$ |
| Chroma CQT | $50.6\% \pm 3.13\%$ |
| PHALAR | $55.2\% \pm 1.78\%$ |

### 4.6.2. LINEAR PROBE FOR CHORDS

We further test PHALAR on a frame-level chord classification task, to verify that it retains harmonic information.

**Method**   We perform a linear probing experiment on GuitarSet (Xi et al., 2018) by training a linear classifier over frozen PHALAR's output embeddings. Specifically, the probe is inserted after the CVNN head, operating on the final complex-valued embeddings $\mathbf{z} \in \mathbb{C}^{512}$ (the same vectors used to compute the bilinear similarity, immediately prior to the weight matrix $\mathbf{W}$). The probe is a complex linear layer mapping $\mathbb{C}^{512} \to \mathbb{C}^{25}$. We take the real part of this output to yield a 25-dimensional real logit vector (one per chord class: Major/Minor$\times$12 keys + No Chord), which is optimized via a standard cross-entropy loss. We evaluate this using a 5-fold cross-validation split by song-ID and compare it against librosa's Chroma CQT baseline, for which we compute the same linear-probe training.

**Results**   In Table 6 PHALAR's embeddings outperform the Chroma CQT ones, suggesting that our architecture's embeddings successfully integrate the harmonic information from the CQT backbone, and map it to a space easier for a linear probe to predict on, allowing it to better resolve harmonic identity.

It should be noted that state-of-the-art systems like `BTC` (Park et al., 2019) achieve $\approx 76\%$ accuracy on chord detection. However, such models utilize deep temporal sequence modeling (e.g., Transformers (Vaswani et al., 2017)) to resolve harmonic ambiguities using long-term context, and predict start and duration of a chord. In contrast, here, without temporal decoding, we use a linear probe on independent frames to predict the simple *presence* of a chord, not the temporal *evolution* of chords in a track.

## 5. Conclusion

We introduce PHALAR, a representation learning framework that replaces learned invariance with enforced equivariance. By leveraging the Fourier Shift Theorem to model temporal alignment as geometric rotation, PHALAR preserves musical coherence discarded by standard pooling and semantic models. It establishes a new state-of-the-art on MoisesDB with a relative improvement of up to $\approx 70\%$ over

COCOLA and significantly higher efficiency. Subjective tests further confirm that PHALAR aligns closer to human perception than industry standards. By addressing the gap where models like CLAP fail to detect temporal misalignment, PHALAR provides a robust, phase-aware metric for evaluating generative audio.

**Limitations and Failure Cases**   Despite its strong performance, PHALAR's reliance on explicit geometric priors introduces specific failure modes:

- **Tempo drift and non-periodic rhythms**: Our Learned Spectral Pooling relies on the Real Fast Fourier Transform (RFFT), which inherently assumes temporal periodicity. As shown in Section G, while PHALAR successfully handles complex non-isochronous meters (e.g., a $7/4$ time signature), its performance degrades when the track undergoes non-periodic tempo changes (e.g., rubato or ritardando). In such cases, phase coherence becomes ill-defined.

- **Arrhythmic and incommensurable strata**: Sustained ambient pads or instruments deliberately operating at unrelated periodicities provide no stable phase reference, limiting the model's ability to lock onto a structural grid.

- **Audio degradation**: As demonstrated in Section D, PHALAR's performance degrades on heavily compressed or lossy audio formats. Aggressive compression can destroy the fine-grained magnitude information in the input spectrogram required to extract reliable phase embeddings.

- **Dataset bias**: Our training distributions heavily feature Western popular music. Consequently, the model's geometric notion of "coherence" may not align with human judgment in contexts where micro-timing deviations are stylistic rather than erroneous.

**Future work**   In future studies, we aim to rigorously test the hypothesis that specific "magnitude-only" feature dimensions represent time-agnostic properties by correlating their phase variance with mood- and key-labeled data. Additionally, we plan to extend this phase-equivariant framework to generative architectures, utilizing complex-valued latents to score generated temporally aligned multitrack audio.

## Impact Statement

This work contributes to the advancement of complex-valued Machine Learning, a field with significant implications for high-dimensional data analysis. By improving signal representation, specifically in Music Information Retrieval, this research offers potential benefits for any domain

relying on phase-sensitive data. This includes non-acoustic fields such as Radar systems, Medical Imaging (MRI), and Time Series Analysis, where preserving the integrity of complex signals is critical for safety and precision.

**Broader impacts and potential misuse**   Within the music domain, PHALAR provides a powerful tool for evaluating and filtering generative audio models. However, when deployed in automated music production workflows or retrieval systems, it risks enforcing rigid, homogenized standards of rhythmic quantization, potentially penalizing stylistic human grooves. Care must be taken to use such metrics as assistive tools rather than absolute arbiters of musical quality.

## Acknowledgements

This work is supported through the MUR FIS2 grant n. FIS-2023-00942 "NEXUS" (cup `B53C25001030001`), and the Sapienza Seed of ERC grant "MINT.AI" (cup `B83C25001040001`). We thank and acknowledge IS-CRA for awarding this project access to the LEONARDO supercomputer, owned by the EuroHPC Joint Undertaking, hosted by CINECA (Italy). We also thank all of the participants in the human evaluation test, and all of the developers that made the games that provided a much needed stress-relief during the creation of this work.

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

# A. Complex-valued layers

Our complex-valued head is designed around phase equivariance, the list of its components is detailed in this section.

## A.1. Complex Linear Layer

The linear layers operate on complex inputs $\mathbf{z} = \mathbf{x} + i\mathbf{y}$ using complex weights $\mathbf{W} = \mathbf{A} + i\mathbf{B}$:

$$\text{CplxLinear}(\mathbf{z}) = (\mathbf{x}\mathbf{A} - \mathbf{y}\mathbf{B}) + i(\mathbf{x}\mathbf{B} + \mathbf{y}\mathbf{A}). \tag{6}$$

By omitting the bias term, this operation commutes with rotation, preserving strict phase equivariance.

## A.2. Complex RMSNorm

Standard normalization methods such as BatchNorm (Ioffe & Szegedy, 2015) and LayerNorm (Ba et al., 2016) rely on mean centering, which disrupts phase relationships. We adopt a complex variant of RMSNorm (Cerovaz et al., 2026) that normalizes based strictly on magnitude:

$$\text{CplxRMSNorm}(\mathbf{z}) = \frac{\mathbf{z}}{\sqrt{\frac{1}{D} \sum_{d=1}^{D} |\mathbf{z}_d|^2 + \epsilon}}, \tag{7}$$

where $|\mathbf{z}_d| = \sqrt{x_d^2 + y_d^2}$ is the magnitude. Since the scaling factor is a real scalar derived from the invariant magnitude, the phase angle of the input is preserved.

## A.3. Complex Activation Functions

We utilize a complex variant of ReLU, modReLU (Arjovsky et al., 2016; Trabelsi et al., 2018; Caragea et al., 2022) which applies a non-linearity to the magnitude while acting as an identity function on the phase:

$$\text{modReLU}(\mathbf{z}) = \frac{\mathbf{z}}{|\mathbf{z}|} \text{ReLU}(|\mathbf{z}| - b) \tag{8}$$

$$= e^{i\angle\mathbf{z}} \cdot \text{ReLU}(|\mathbf{z}| - b). \tag{9}$$

This effectively gates the magnitude of the phasors based on a learned bias $b$, allowing the model to learn non-linear interactions between features while maintaining their temporal alignment.

# B. Comparison with original COCOLA results

In this paper we retrained the COCOLA baseline to ensure fairness when accounting for the optimization algorithm shift (Adam (Kingma, 2014) to Muon (Jordan et al., 2024)) that we take with respect to (Ciranni et al., 2025). In Table 7 we present the original results from (Ciranni et al., 2025) next to PHALAR and our retrained COCOLA baseline.

# C. MERT Cross-Architecture comparison

In the main text (Table 1), we compared PHALAR against frozen MERT embeddings (m-a-p/MERT-v1-95M) expanded with our Learned Spectral Pooling and CVNN head. To fully validate the necessity of this phase-aware architecture, we conducted an ablation over MERT aggregation strategies, evaluating three configurations trained under the exact same regime described in Section 4.1:

1. MERT-freeze: Global Average Pooling + cosine similarity (representing off-the-shelf semantic features).

2. MERT-avg: Global Average Pooling + trainable real-valued MLP head + bilinear similarity.

3. MERT-cplx (the variant shown in Table 1): Learned Spectral Pooling + trainable CVNN head + complex bilinear similarity (the PHALAR head).

As expected, in Table 8 MERT-freeze fails to solve the task, mirroring the collapse to random chance seen in CLAP and CDPAM. This reinforces the observation that raw semantic embeddings invariant to temporal structure cannot assess

*Table 7.* **Contrastive retrieval with original COCOLA results** (↑) Top-1 accuracy on disjoint submix retrieval. (†: reported in (Ciranni et al., 2025))

| Dataset | K | PHALAR (2.3M) | COCOLA† (5.2M) | COCOLA (5.2M) |
|---|---|---|---|---|
| MoisesDB | 8 | **86.79** | 73.68 | 75.81 |
| MoisesDB | 16 | **81.49** | 62.17 | 64.44 |
| MoisesDB | 64 | **70.87** | 34.04 | 41.84 |
| Slakh2100 | 8 | **87.69** | 79.72 | 79.33 |
| Slakh2100 | 16 | **83.28** | 72.62 | 71.58 |
| Slakh2100 | 64 | **72.37** | 59.35 | 55.84 |
| ChocoChorales | 8 | **99.65** | 98.27 | 97.82 |
| ChocoChorales | 16 | **99.45** | 96.67 | 96.02 |
| ChocoChorales | 64 | **98.61** | 90.67 | 89.34 |

*Table 8.* **Contrastive retrieval ablation on MERT** (↑) Top-1 accuracy on disjoint submix retrieval for different aggregation and projection heads on frozen MERT embeddings.

| Dataset | K | MERT-freeze | MERT-avg | MERT-cplx |
|---|---|---|---|---|
| MoisesDB | 8 | 14.06 | 63.53 | 67.39 |
| MoisesDB | 16 | 7.63 | 50.58 | 59.13 |
| MoisesDB | 64 | 1.83 | 27.82 | 45.85 |
| Slakh2100 | 8 | 15.77 | 63.81 | 66.70 |
| Slakh2100 | 16 | 8.99 | 52.41 | 58.39 |
| Slakh2100 | 64 | 3.35 | 32.64 | 46.13 |
| ChocoChorales | 8 | 6.44 | 92.36 | 96.49 |
| ChocoChorales | 16 | 2.37 | 86.41 | 93.79 |
| ChocoChorales | 64 | 0.31 | 68.74 | 86.65 |

coherence. Introducing a trainable real-valued projection head (`MERT-avg`) extracts some latent structural information, drastically improving performance. However, replacing Global Average Pooling with our Learned Spectral Pooling and CVNN head (`MERT-cplx`) provides a further significant boost across all datasets. This confirms that even for large-scale semantic foundation models, explicit phase-equivariant processing is the optimal strategy for resolving musical coherence.

## D. Audio Degradation Correlation

We evaluate how PHALAR correlates with audio degradation by reconstructing full-mixture excerpts from MUSDB using two neural audio codecs, **DAC** (Kumar et al., 2023) and **EnCodec** (Défossez et al.), at varying codebook depths ($K$). Such that lower $K$ should result in significant information loss and audio degradation.

As shown in Table 9, all models (except for $\text{FAD}_{\text{MERT}}$) demonstrate a monotonic relationship with audio quality. Proving the intuitive notion that both Semantic Similarity and Structural Coherence tasks correlate with audio quality. Confirming that PHALAR's phase-aware objective successfully captures **structural fidelity**.

## E. Theoretical Bounds of the Coherence Metric

While the standard Cosine Similarity is strictly bounded to $[-1, 1]$, our Equation (3) is effectively a generalized inner product. A potential concern is that this score is unbounded. However, since our embeddings are L2-normalized ($||\mathbf{z}||_2 = 1$), the metric space is strictly bounded by the spectral properties of the weight matrix $\mathbf{W}$.

*Table 9.* **Codebook Ablation Test.** Comparison of metric scores on audio reconstructed via neural codecs (DAC, Encodec) at different codebook counts ($K$). Higher $K$ corresponds to higher audio quality.

| Condition | PHALAR ($\uparrow$) | CLAP ($\uparrow$) | CDPAM ($\downarrow$) | FAD$_{\text{MERT}}$ ($\downarrow$) | FAD$_{\text{CLAP}}$ ($\downarrow$) | Audiobox$_{\text{PQ}}$ ($\uparrow$) |
|---|---|---|---|---|---|---|
| *DAC* ($K = 1$) | 8.310 | 0.718 | 0.140 | 21.85 | 0.591 | 6.52 |
| *DAC* ($K = 3$) | 9.515 | 0.890 | 0.075 | 21.88 | 0.229 | 7.43 |
| *DAC* ($K = 6$) | 9.844 | 0.940 | 0.042 | 21.88 | 0.146 | 7.72 |
| *DAC* ($K = 9$) | 9.974 | 0.960 | 0.033 | 21.90 | 0.105 | 7.76 |
| *EnCodec* ($K = 2$) | 9.513 | 0.893 | 0.067 | 21.85 | 0.387 | 7.38 |
| *EnCodec* ($K = 4$) | 9.752 | 0.929 | 0.050 | 21.87 | 0.356 | 7.65 |
| *EnCodec* ($K = 8$) | 9.954 | 0.955 | 0.038 | 21.84 | 0.302 | 7.72 |
| *EnCodec* ($K = 16$) | 10.035 | 0.972 | 0.031 | 21.84 | 0.236 | 7.76 |

### E.1. General Bound via Singular Values

For the asymmetric scoring function used during training, the score is bounded by the spectral norm of $\mathbf{W}$, which is equivalent to its largest singular value $\sigma_{\max}$

$$|s(\mathbf{z}_x, \mathbf{z}_y)| \leq |\mathbf{z}_x^H \mathbf{W} \mathbf{z}_y| \leq ||\mathbf{W}||_2 = \sigma_{\max}(\mathbf{W}), \tag{10}$$

implying that $\sigma_{\max}$ acts as a learnable temperature parameter for the InfoNCE loss. If a fixed range is required, $\mathbf{W}$ can be spectrally normalized (Miyato et al., 2018) at inference time.

### E.2. Tighter bound via symmetrization

In our evaluation we utilize Equation (4), symmetrizing the coherence score. As we argue in Section 4.4, it is equivalent to replacing $\mathbf{W}$ with its Hermitian part. Unlike $\mathbf{W}$, the matrix $\mathbf{W}_{\text{eff}}$ is Hermitian, and consequently, by the spectral theorem, all its eigenvalues $\lambda_i$ are guaranteed to be real numbers (Cauchy, 1829). This allows us to bound the symmetrized score strictly by the eigenvalues of $\mathbf{W}_{\text{eff}}$

$$|s_{\text{comm}}(\mathbf{z}_x, \mathbf{z}_y)| \leq \max |\lambda(\mathbf{W}_{\text{eff}})|. \tag{11}$$

Since $\max |\lambda(\mathbf{W}_{\text{eff}})| \leq \sigma_{\max}(\mathbf{W})$, the symmetrized metric can be provided with a tighter bound than the raw score, effectively filtering-out the skew-Hermitian energy that does not contribute to the real-valued coherence.

## F. Direct Test of Phase-Aware Behavior

A core claim of PHALAR is that temporal alignment is explicitly encoded as a phase rotation in the complex latent space. To directly test this behavior beyond downstream retrieval metrics, we designed an experiment to empirically verify whether the Fourier Shift Theorem operates linearly within the model's embeddings.

**Method** We extracted embeddings from the first four bass bars of "Seven Nation Army" (White, 2003). We systematically varied the temporal offset of the audio by applying a shifting delay $\Delta t \in [0, 1.7s]$ via zero-padding at the beginning of the track. For each shifted input, we extracted the complex-valued embedding $\mathbf{z}$ and measured the Pearson correlation $\rho$ between the applied delay $\Delta t$ and the unwrapped phase angle of each latent dimension.

**Result** Looking at the individual dimensions with the highest absolute correlation (Figure 6), we observed clear linear relationships, with feature phases reliably increasing or decreasing (positive and negative slopes) in direct proportion to $\Delta t$.

To quantify the global behavior of the embedding, we normalized the directions by inverting the negative slopes and computed the magnitude-weighted average of the unwrapped phase across all dimensions. This weighted average phase grows perfectly linearly with the temporal delay, yielding a Pearson correlation of $\rho \approx 0.999$.

This confirms that PHALAR's architecture successfully translates time shifts in the input domain into geometric phase rotations in the latent space, empirically proving that it preserves temporal alignment as a mathematical primitive.

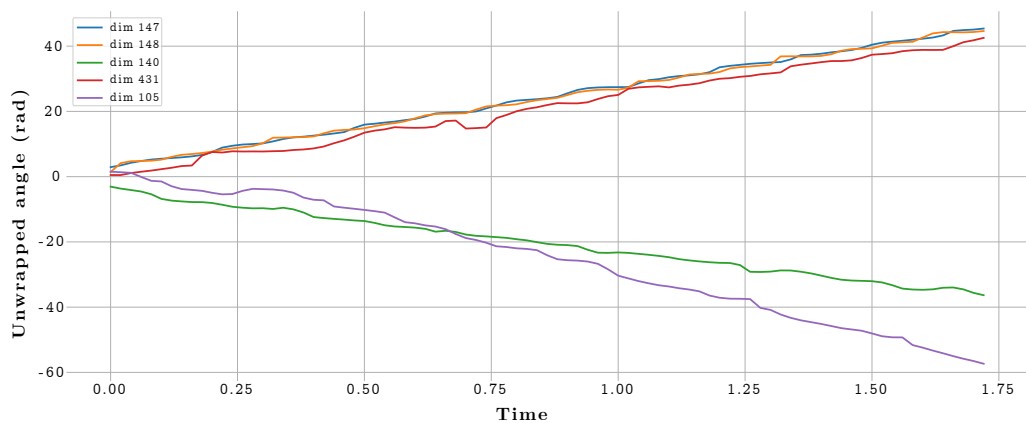

*Figure 6.* $\Delta t$ v. $\angle \mathbf{z}$ over the top-5 dimensions by $|\rho|$

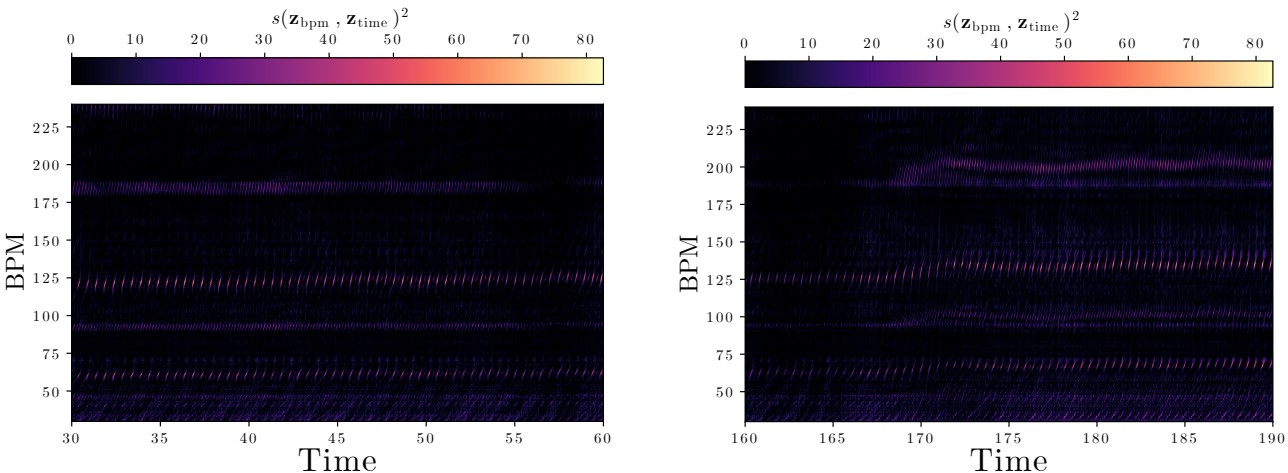

*Figure 7.* BPM of "Money" (Waters, 1973)

## G. Behavior Under Non-Isochronous Rhythms and Tempo Drift

In Section 5, we noted that PHALAR's Learned Spectral Pooling relies on the RFFT, which assumes temporal periodicity. To investigate how this assumption impacts real-world music, we evaluated PHALAR's zero-shot beat tracking capabilities (using the synthetic metronome probe described in Section 4.6.1) on tracks with non-isochronous rhythms and dynamic tempos.

**Complex Meters (Non-Isochronous Rhythms)** We tested the model on "Money" (Waters, 1973), a track famous for its $^7/_4$ time signature at 126 BPM. Because a $^7/_4$ meter is still fundamentally periodic (repeating every 7 beats), PHALAR gracefully handles the non-isochronous feel. As shown in Figure 7, the zero-shot probe successfully recovers the underlying pulse at the correct 126 BPM, generating clear interference patterns in the similarity matrix. This proves the model is not biased toward standard $^4/_4$ structures, but rather detects true rhythmic periodicity.

**Tempo Drift** However, as theoretically expected, the model is significantly less reliable when the beat itself accelerates or decelerates non-periodically (tempo drift). In the same track ("Money"), the band switches to a standard $^4/_4$ signature around the 172-second mark for the guitar solo, introducing a distinct change of pace. During this transition, the horizontal bands in the similarity heatmap become blurred and unstable (Figure 7). Because the tempo fluctuates, the phase coherence of the rhythmic grid becomes ill-defined, preventing the RFFT from locking onto a single stable frequency.

This confirms that PHALAR is highly robust to complex metrical structures provided they are periodic, but its performance predictably degrades in the presence of human tempo drift or rubato.