# OpenReview forum: "PHALAR: Phasors for Learned Musical Audio Representations"
_ICML.cc/2026/Conference — ICML 2026 regular_

### Official Review · Reviewer_sxzi · 2026-03-07

**Soundness:** 2
**Presentation:** 3
**Significance:** 3
**Originality:** 3
**Overall Recommendation:** 4
**Confidence:** 4

**Summary:**

This paper proposes a contrastive framework for music audio representation learning which enforces pitch invariances and phase equivariances, which is suitable for coherence tasks. The framework is evaluated on a new coherence setup where it outperforms other methods, and its applicability is also tested on other MIR downstream tasks.

**Compliance With Llm Reviewing Policy:**

Affirmed.

**Final Justification:**

This paper is overall well written using a technically appropriate methodology. I had certain concerns in my original review regarding claims made or certain comparisons in the experimental section, which were addressed by the authors in the rebuttal. Overall this is a solid paper and I recommend to maintain the overall recommendation score to a 4.

**Key Questions For Authors:**

No specific questions to ask the authors.

**Limitations:**

yes

**Strengths And Weaknesses:**

Strengths:
* Good motivation and justification towards a phase-equivariant representation learning framework for coherence tasks.
* Well written paper, clear structure and good use of maths and figures throughout.
* Technically appropriate methodology and network architecture.

Weaknesses:
* Certain comparisons with baseline models are not fully valid, since those models were designed for similarity tasks, not coherence tasks. From what I can see, only COCOLA can be deemed as a valid and fair comparison for coherence evaluations.
* It is unclear why the proposed model needs to be pitch-invariant. Would a pitch-equivariant model be more suitable in order to perform harmonic coherence? The motivation on this part is unclear, and perhaps incorrect. This needs to be thoroughly justified and explained, and relevant limitations need to be included in the final section.
* Section 5 is not useful to the reader, since it does not include any critical discussion on potential limitations of the current proposed framework and does not provide any areas for future work. This would need to be radically expanded and rewritten.

---

> ### Author Rebuttal · Authors · 2026-03-30
>
> We thank the reviewer for the constructive critique and in particular for the precise observation on pitch equivariance.
>
> ### CLAP and CDPAM comparisons, W1
>
> > *Certain comparisons with baseline models are not fully valid, since those models were designed for similarity tasks, not coherence tasks. From what I can see, only COCOLA can be deemed as a valid and fair comparison for coherence evaluations.*
> >
>
> We agree that CLAP and CDPAM are introduced here strictly as diagnostic probes. Their near-random performance on stem retrieval is not an indictment of their design; rather, it is controlled evidence that semantic similarity and structural coherence require categorically different geometric inductive biases, and that strong performance on one does not transfer to the other.
>
> To the best of our knowledge, COCOLA is the only fair coherence baseline (though we would gladly add others if the reviewer has suggestions), and our improvement over it constitutes the paper's primary empirical claim. In the revision, we will more strongly emphasize their introduction as probes rather than competitors. Furthermore, we will expand our analysis by including the MERT tests detailed in our response to Q2 for Reviewer `zxvh`. These tests provide empirical evidence of PHALAR’s performance compared to off-the-shelf solutions applied to a frozen pre-trained MERT model.
>
> ### Pitch-Invariance v. -Equivariance, W2
>
> > *It is unclear why the proposed model needs to be pitch-invariant. Would a pitch-equivariant model be more suitable in order to perform harmonic coherence? […] This needs to be thoroughly justified and explained, and relevant limitations need to be included in the final section.*
> >
>
> We thank the reviewer for this precise terminological observation. They are correct: *pitch-equivariance* is the technically accurate description of what our backbone provides, and we will update the manuscript accordingly.
>
> Our use of the term "pitch-invariance" referred to the inductive bias of the convolutional kernels operating on the CQT. A convolutional kernel working on CQT data encounters the same frequency-interval pattern (e.g., a "major third") at the same relative bin offset regardless of the absolute pitch. It recognizes intervals without needing separate parameters per key (and is thus invariant to the absolute pitch). By contrast, on a Mel-spectrogram, the spectral shape of the same interval varies across octaves due to filter-bank resolution differences, forcing the network to learn redundant, key-specific representations. This is the intuition we intended to convey, but from a feature-map perspective, it is correct to describe the backbone as pitch-equivariant (i.e., shifting the input by $k$ bins will result in an equivalent $k$-shift in the feature map).
>
> Regarding the reviewer's deeper concern, we agree: harmonic coherence requires pitch-awareness, not pitch-invariance. The architecture explicitly handles this in two stages:
>
> 1. Equivariant backbone: The axial CNN efficiently extracts high-quality, interval-aware harmonic features without wasting capacity on key-specific redundancies.
> 2. Pitch-aware projection: In Eq. 1, the backbone features are projected onto the learned basis $\mathbf{W}\_\textrm{proj}$, pointwise in time. This projections operates simultaneously over *all* frequency bins, meaning the resulting embedding $\mathbf{Z}\_\textrm{time}$ encodes both the harmonic interval structure (from the backbone) and the absolute frequency position of those intervals (from the full-frequency projection). The model therefore becomes pitch-aware at the point of spectral pooling.
>
> The necessity for the CQT (and thus the backbone's pitch equivariance) is confirmed in Table 4. Replacing the CQT with a Mel-spectrogram reduces accuracy by $1.66\%$, demonstrating that the equivariant backbone improves harmonic extraction efficiency even though the final task requires pitch awareness. We will ensure the terminology is corrected and this justification is clearly explained throughout the revised manuscript.
>
> ### Conclusions need to be expanded, W3
>
> > *Section 5 is not useful to the reader, since it does not include any critical discussion on potential limitations of the current proposed framework and does not provide any areas for future work. This would need to be radically expanded and rewritten.*
> >
>
> We agree entirely. We will radically expand Section 5 to include technical limitations, scope, tempo stability assumptions (as described in our response to Q5 for Reviewer `u65s`), dataset bias, potential cases of misuse, and directions for future work. We will use the extra page allowed in the final revision to accommodate these additions.

---

> > ### Author Rebuttal · Reviewer_sxzi · 2026-03-31
> >
> > The authors have successfully addressed my comments, and in parcicular the clarification regarding pitch-equivariance / pitch-invariance is welcome.

---

> > > ### Author Response · Authors · 2026-04-01
> > >
> > > We thank the reviewer for the positive reassessment of our work and for the constructive feedback throughout the review process. We are glad that our clarifications, particularly regarding pitch equivariance vs. invariance, have addressed the concerns.
> > >
> > > We will incorporate all discussed improvements in the final version to further strengthen the paper.
> > >
> > > We remain available for any additional questions or clarifications. If the reviewer feels that their concerns have been fully resolved, we would kindly invite them to consider updating their score to reflect their current assessment.

---

### Official Review · Reviewer_r7tm · 2026-03-10

**Soundness:** 4
**Presentation:** 3
**Significance:** 3
**Originality:** 3
**Overall Recommendation:** 6
**Confidence:** 4

**Summary:**

The authors propose PHALAR, a contrastive neural network framework for the task of music stem retrieval. Stem retrieval is a complicated task as it requires understanding of both tempo, harmonic content, and texture. The complex-valued neural network head introduced in PHALAR, as well as the symmetric bilinear similarity score, improves stem retrieval performance, can predict tempo in a zero-shot framework, and also captures chord information.

**Compliance With Llm Reviewing Policy:**

Affirmed.

**Final Justification:**

The authors expanded the scope of their evaluation and addressed my remaining concerns.

**Key Questions For Authors:**

How is random cropping performed to not disturb the beat structure of the stems?
Where is the linear probe inserted for the chord estimation task?

**Limitations:**

The primary limitation of this paper is not comparing PHALAR to more relevant music understanding models like MERT. Despite this, I believe the conceptual contribution is significant.

**Strengths And Weaknesses:**

Soundness:

This papers makes sound arguments regarding the loss of fine-grained temporal information when using global average pooling and how the use of an RFFT on internal representations can maintain said information via complex phase information. The contrastive retrieval benchmark makes a strong case for choosing PHALAR over COCOLA, though relevant music understanding models like MERT (Li 2023) are not included. The human-centric validation is much appreciated, including its investigation of correlation with human perception and corrections via linear mixed effects analysis. The ablation study is well motivated and makes a strong case for the benefit of phase equivariance.

A small question about the random cropping augmentation - how is this performed to not disturb the beat structure of the stems? Does this happen from the start of the stem up until T seconds, with the rest padded with silence?

Presentation:

The paper is presented clearly, with a few points worth clarifying.

One issue: The authors “posit” in section 4.5 that “magnitude-only features emerge to represent time-agnositc qualities,” but do not present evidence to support this. Their examination of “emergent structures” does not divide analysis into individual feature dimensions with the characteristics they highlight in Figure 4. This is a “conjecture” that requires further, rigorous experimentation.

In addition: It is unclear to me where the linear probe is inserted in section 4.6.2. Is this a complex linear probe operating on $z_{X}$ in Figure 2? Or a real valued probe on $z_{time}$?

Figure 3 needs more explanation, at the moment it is confusing. PHALAR appears to align better to human preference for higher rated model scores than lower, but the other models appear to align better to human preference for lower model scores.


Significance:

The paper contributes to the growing field of stem retrieval, and is likely directly applicable to stem generation tasks as well.


Originality:

This is an original approach to the best of my knowledge.

---

> ### Author Rebuttal · Authors · 2026-03-30
>
> We thank the reviewer for their positive assessment and recommendation for acceptance. We address their clarifying questions below:
>
> ### Random cropping and beat structure, Q1
>
> > *How is random cropping performed to not disturb the beat structure of the stems?*
> >
>
> The same temporal offset is applied to both the anchor and positive submixes simultaneously. Both are cropped at the exact same position in time, so their relative beat alignment is fully preserved. Thus, the rhythmic relationship between the anchor and the positive remains identical to the original track. What varies across training examples is simply the absolute temporal position of the crop within the track.
>
> ### Chord estimation’s linear probe, Q2
>
> > *Where is the linear probe inserted for the chord estimation task?*
> >
>
> The linear probe operates on the final complex-valued embeddings $\mathbf{z}\in\mathbb{C}^{512}$ output by the CVNN head (these are the same embeddings used to compute the bilinear similarity score, immediately before the weight matrix $\mathbf{W}$). The probe is a complex linear layer (Eq. 6) mapping $\mathbb{C}^{512}\to\mathbb{C}^{25}$. We then take the real part of the output to yield a $25$-dimensional real logit vector (one per chord class: Major/Minor $\times$ 12 keys + No Chord). The probe is trained on frozen PHALAR embeddings using a cross-entropy loss. We will clarify this in the text.
>
> ### Magnitude-only features conjecture, W1
>
> > *The authors “posit” in section 4.5 that “magnitude-only features emerge to represent time-agnositc qualities,” but do not present evidence to support this. […] This is a “conjecture” that requires further, rigorous experimentation.*
> >
>
> The use of the word "posit" was intentional. In the revision, we will make the speculative nature of this claim even clearer by explicitly labeling it as a "conjecture" in the section heading. We will also note that rigorous confirmation would require correlating the phase variance of individual feature dimensions with key- and mood-labeled data, which we leave for future work.
>
> ### Figure 3 LOWESS regression confusion, W2
>
> > *Figure 3 needs more explanation, at the moment it is confusing. PHALAR appears to align better to human preference for higher rated model scores than lower, but the other models appear to align better to human preference for lower model scores.*
> >
>
> We thank the reviewer for pointing out this presentation issue. We will replace Figure 3 with a set of more informative quintile visualizations in the revision (`rfig5-7` at [https://anonymous.4open.science/r/icml_rebuttals_images_phalar-006E](https://anonymous.4open.science/r/icml_rebuttals_images_phalar-006E)).
>
> LOWESS regression can be sensitive to sparse data at the tails of each model's score distribution. When the low-score regime contains few data points that are themselves noisy, the fit curve flattens or reverses, creating the visual impression that the baselines align better at the low end. This is a distributional artifact of the visualization method, not a property of the model.
>
> To address this directly, we provide quintile$\times$quintile binned rank heatmaps for PHALAR (`rfig5`), Audiobox CE (`rfig6`), and COCOLA (`rfig7`), where the key diagnostic is performance at the distribution's extremes.
>
> - **At the low end** (Q1 human ratings), the fraction of samples assigned to the bottom two model quintiles (Q1-Q2) is: PHALAR $71\%$ (50/70), Audiobox CE $61\%$ (43/70), COCOLA $53\%$ (37/70). All three models detect incoherent stems, but PHALAR performs best.
> - **At the high end** (Q5 human ratings), the fraction of samples assigned to the top two model quintiles (Q4-Q5) is: PHALAR $71\%$ (49/69), Audiobox CE $54\%$ (37/69), COCOLA $43\%$ (30/69). Again, PHALAR performs best here, while Audiobox and COCOLA underperform (as Figure 3 also shows).
>
> Thus, PHALAR discriminates equally well at both ends of the coherence spectrum. Figure 3 will be replaced to better communicate this comparison.
>
> ### More relevant comparison, L1
>
> > The primary limitation of this paper is not comparing PHALAR to more relevant music understanding models like MERT.
> >
>
> As detailed in our response to Q2 for Reviewer `zxvh`, we ran a number of tests on a frozen pre-trained MERT model, using both Global Average Pooling and our learned Spectral Pooling Layer. All configurations underperformed compared to PHALAR. We will include these results in the final revision to provide a more comprehensive analysis.

---

> > ### Author Rebuttal · Reviewer_r7tm · 2026-04-03
> >
> > The authors have completely addressed my concerns. The thorough examination of MERT strengthens the case for their proposed model. I have raised my rating accordingly.

---

> > > ### Author Response · Authors · 2026-04-04
> > >
> > > We sincerely thank the reviewer for their highly constructive feedback and for their willingness to reassess our work and raise the score further. Your suggestion to evaluate against the MERT baseline was particularly valuable not only for our discussion with you, but also for addressing questions from the other reviewers. We are also glad that our updated visualizations successfully addressed your concerns, and will ensure all these additions are incorporated into the final manuscript. Thank you again for your time and review!

---

### Official Review · Reviewer_u65s · 2026-03-12

**Soundness:** 3
**Presentation:** 3
**Significance:** 2
**Originality:** 2
**Overall Recommendation:** 4
**Confidence:** 4

**Summary:**

This paper proposes PHALAR, a contrastive representation learning framework for musical audio that is designed to preserve temporal alignment information, which is often lost in standard pooling-based models. The method combines a real-valued feature extractor with a learned spectral pooling module and a complex-valued projection head, aiming to model rhythmic coherence through phase-equivariant representations. The paper focuses on stem-to-mix retrieval and reports strong improvements over prior methods on multiple datasets, while also showing better efficiency in terms of model size and training cost. In addition, the authors provide supporting experiments suggesting that the learned representation captures broader musical structure beyond the main retrieval task.

**Compliance With Llm Reviewing Policy:**

Affirmed.

**Final Justification:**

The rebuttal addressed my main concerns. The new phase-time linearity experiment and the gain decomposition (10.1 from backbone/optimizer vs. 18.9 from spectral pooling and CVNN head) clarify the source of improvements. The MERT cross-architecture comparison further supports the contribution of the phase-aware mechanism. I raise my score from 3 to 4. I expect the promised revisions to Section 3 and Section 5 to appear in the final manuscript.

**Key Questions For Authors:**

1. The main claim is that PHALAR captures temporal alignment through phase aware representations. Could the authors provide a more direct test of this claim, beyond Figures 1 and 4 and the retrieval improvements? A stronger answer here would increase my confidence in the core mechanism.
2. The introduction currently moves quite quickly from motivation to method. Could the authors state more clearly what the exact research problem is, what the main challenge is, and how this differs from prior music representation learning work that mainly targets semantic similarity? This would improve my assessment of the paper’s positioning.
3. The method section is very short for a paper centered on learned spectral pooling, the CVNN head, and the Hermitian bilinear score. Could the authors clarify the main implementation details and key hyperparameters needed to understand and reproduce the method? A clear response would improve both soundness and presentation.
4. The ablations suggest that spectral pooling and the complex valued design matter, but it is still not fully clear how much of the gain comes from the claimed phase based mechanism itself versus the overall architecture and training setup. Could the authors comment on this more directly? A convincing answer would strengthen my view of the technical contribution.
5. Could the authors discuss the main failure cases of PHALAR, and in what kinds of music settings the proposed phase based inductive bias may be less effective? This would help me better judge the scope and robustness of the method.

**Limitations:**

No. The paper would benefit from a brief discussion of its practical limitations, such as its focus on stem to mix retrieval, the domain specificity of the phase based inductive bias, and cases where temporal coherence may not align with human musical judgment. A short note on broader use in generative or retrieval systems would also be helpful, including possible misuse in music production workflows and bias from dataset composition.

**Strengths And Weaknesses:**

Soundness: The paper is reasonably sound, and the empirical results are fairly strong, especially on the stem to mix retrieval task across Slakh2100, MUSDB18, and MoisesDB. The ablation results are also useful, because they show that removing the spectral pooling or replacing the complex valued design leads to worse performance, which suggests that the gains are not coming from the backbone alone. A remaining issue is that the central claim is still somewhat stronger than the evidence provided. The evidence comes mostly from the Fourier shift intuition, Figures 1 and 4, and downstream gains, not from a direct test of the claimed phase aware behavior. The method section is also quite short, so some important details of the learned spectral pooling, CVNN head, and Hermitian bilinear score are still not clear enough. Overall, the paper looks technically plausible and experimentally supported, but the core mechanism is not yet fully established.

Presentation: The paper is generally readable, and the overall idea can be followed without too much difficulty. The main presentation issue is in the introduction section, which moves quite quickly from the motivation to PHALAR, but gives too little discussion of the most relevant prior work and does not clearly state the concrete research problem and challenge. As a result, the method can feel a bit solution driven. Figure 1 is helpful for intuition, but it works more as a conceptual illustration than a strong explanation of the method. The method section is also too brief for a paper whose main contribution lies in learned spectral pooling, the complex valued head, and the bilinear scoring design. Some key technical details are therefore hard to recover from the main text. Overall, the paper is understandable, but the positioning and technical exposition should be improved.

Significance: The paper addresses a meaningful problem in music representation learning. Instead of treating the task as standard audio similarity learning, it focuses on stem to mix coherence, where the key question is whether a representation can capture relative temporal fit rather than just shared content. This is a worthwhile problem, since in music applications a stem can be semantically similar to a mix but still be a poor fit in rhythm or structure. In that sense, the paper pushes the discussion beyond generic embedding quality. The significance is mostly within music and audio ML rather than broad across all of machine learning, but that scope is appropriate here. If the main idea holds up, PHALAR could be useful not only for retrieval, but also for other tasks where alignment and coherence matter, including music generation and arrangement. The paper is therefore solving a community relevant problem, and its main significance comes from highlighting temporal coherence as a representation learning target, not just from improving a benchmark.

Originality: The paper has a reasonable level of originality, though not mainly because of a completely new model component. The backbone, spectral transform, and complex valued processing are not individually new ideas. The more original part is how the paper puts them together around a specific claim, namely that stem to mix coherence should be modeled through phase aware representations that preserve relative temporal alignment. In that sense, the novelty comes more from the problem formulation and the design logic of PHALAR than from any single architectural block. I do not view it as a simple module stacking paper, because the temporal spectral pooling, the complex valued head, and the Hermitian bilinear score are all tied to the same phase based motivation. Still, the paper should do a better job of distinguishing this contribution from related work in music representation learning and complex valued audio models, especially in the introduction, so that the originality is stated more sharply.

---

> ### Author Rebuttal · Authors · 2026-03-30
>
> We thank the reviewer for their thorough and constructive critique. We address each point directly below.
>
> ### Direct test of phase-aware behavior, Q1
> > *PHALAR captures temporal alignment through phase aware representations. Could the authors provide a more direct test of this claim?*
>
> We designed a new experiment (figures: https://anonymous.4open.science/r/icml_rebuttals_images_phalar-006E) showing PHALAR embeddings exhibit a linear phase-time relationship, confirming temporal shifts are encoded as phase rotations.
>
> We extract embeddings from the first four bass bars of "Seven Nation Army" while varying the temporal offset by $\Delta t \in [0, 1.7\textrm{s}]$ via padding, measuring Pearson correlation between the unwrapped phase and $\Delta t$ per dimension.
>
> `rfig1` shows per-dimension behavior: top-5 dimensions by $|\rho|$ have both positive and negative slopes (phase increases/decreases with delay). In `rfig2`, we normalize directions by inverting the negative slopes and compute the magnitude-weighted average unwrapped phase, which grows linearly with the delay ($\rho\sim0.999$), confirming the Fourier Shift Theorem operates in latent space.
>
> We will include this experiment as an appendix in the final revision.
>
> ### Research problem definition, Q2
> > *Could the authors state what the research problem is [...] and how this differs from prior music representation learning work?*
>
> Our goal is to model *musical coherence*: given a partial mix (e.g., drums + bass), identify which stems fit with it. This differs from standard semantic tasks, where the objective is to recognize *what* is present (e.g., “guitar”), regardless of timing.
>
> The key challenge is that coherence depends on *temporal alignment*. Two signals can contain the same instruments but be incoherent if they are misaligned (e.g., drums slightly off-beat with the bass). Existing models typically use global average pooling (GAP), which removes temporal ordering and maps these different cases to similar embeddings. As a result, prior work (e.g., CLAP, COCOLA) discards precisely the information needed for this task.
>
> Our contribution is an aggregation scheme that preserves temporal alignment by construction, enabling the model to distinguish between coherent and incoherent combinations. We will revise Section 1 to present this distinction more explicitly.
>
> ### Implementation details, Q3
> > *Could the authors clarify the main implementation details and key hyperparameters?*
>
> We will expand Section 3 with the following details:
>
> - Learned spectral pooling: The projection matrix $\mathbf{W}_\textrm{proj}$ maps the flattened backbone features to $D=80$ dimensions. After applying the RFFT (Eq. 2) with $C=8$, each embedding contains $D\times C=640$ complex values. Matching COCOLA's $1280$ real values.
> - CVNN head: This consists of two complex linear layers. The first is followed by $\textrm{CplxRMSNorm}$ and $\textrm{modReLU}$. The output-dimension is $512$ complex values. The bilinear matrix maintains the same dimensionality of $512\times512$.
>
> ### Phase mechanism v. Overall architecture gains, Q4
> > *It is not fully clear how much of the gain comes from the architecture and training setup. Could the authors comment on this more directly?*
>
> The majority of gains ($\sim19$ points) come specifically from spectral pooling and phase-aware processing, not the backbone or the optimizer. The ablation in Table 4 allows for a precise decomposition.
>
> The configuration *w/o Spectral Pooling (Global Avg Pool + Real MLP)* gives an "approximation-only, no-phase" baseline: it uses our backbone and the Muon optimizer, but replaces the spectral head with standard GAP and a real-valued MLP. This achieves $51.97\%$, which is already $10.1$ points above the retrained COCOLA baseline ($41.84\%$). This gap isolates gains attributable to the axial backbone and optimizer upgrade, independent of any phase mechanism.
>
> The additional $18.9$ points required to reach the full PHALAR model come from replacing GAP with Learned Spectral Pooling and the real MLP with the CVNN head, as confirmed by the ablations in Table 4.
>
> ### Main failure cases, Q5
> > *Could the authors discuss the main failure cases of PHALAR?*
>
> We expect degraded performance in at least three scenarios:
>
> 1. Tempo drift: The RFFT assumes temporal periodicity. Music with significant tempo fluctuations violates this assumption; phase coherence becomes ill-defined when the beat accelerates or decelerates non-periodically. Our answer to Q1 for Reviewer `zxvh` shows an example of this.
> 2. Arrhythmic or incommensurable rhythmic strata: sustained pads or instruments deliberately operating at unrelated periodicities provide no stable phase reference.
> 3. Lossy or compressed audio: As shown in Appendix C (Table 8), PHALAR's performance degrades with quality. Aggressive compression can degrade information in the input spectrogram, reducing the quality of the embeddings.
>
> We will expand Section 5 to discuss these limitations.

---

> > ### Author Rebuttal · Reviewer_u65s · 2026-04-03
> >
> > The authors have addressed my concerns with concrete new experiments and clear gain decomposition. I raise my score to 4.

---

> > > ### Author Response · Authors · 2026-04-03
> > >
> > > We sincerely thank the reviewer for their rigorous and constructive feedback, and especially for their willingness to reassess our work and raise the score. Your prompt to provide a direct test of the phase-aware behavior was very valuable; it allowed us to empirically validate our core mechanism and significantly strengthened the paper. We are very grateful for the time and effort you dedicated to improving our submission, and we will incorporate all these additions into the final manuscript.

---

### Official Review · Reviewer_zxvh · 2026-03-12

**Soundness:** 3
**Presentation:** 3
**Significance:** 3
**Originality:** 2
**Overall Recommendation:** 4
**Confidence:** 4

**Summary:**

This paper introduces PHALAR (Phasors for Learned Musical Audio Representations), a contrastive learning framework based on complex-valued phasors and phase equivariance. Through the introduction of a learned spectral pooling layer and a complex-valued projection head, it explicitly maps the temporal alignment features of audio to phase rotations in the frequency domain. This model addresses the issue where existing audio representation models discard temporal information due to an over-pursuit of translational invariance, which makes them unable to effectively handle structural coherence tasks such as rhythmic and harmonic alignment in stem retrieval. Experimental results show that PHALAR achieves a 70% relative increase in retrieval accuracy compared to current state-of-the-art models, while reducing the parameter count by more than half and increasing training speed by 7 times.

**Compliance With Llm Reviewing Policy:**

Affirmed.

**Key Questions For Authors:**

In the zero-shot beat tracking experiment, the model utilizes a synthetic metronome probe to identify periodic interference patterns . How does this mechanism perform on music with significant timing fluctuations, such as non-isochronous rhythms?

The reported 70% relative increase in accuracy is primarily measured against COCOLA and semantic models like CLAP. Since CLAP is fundamentally designed for semantic invariance and is expected to fail at structural tasks , have the authors considered benchmarking against other time-sensitive or phase-aware architectures used in source separation?

**Limitations:**

yes

**Strengths And Weaknesses:**

This paper is clear and exceptionally well-structured.

This work provides a vital insight into the blindness of current large-scale foundation models like CLAP, which perform at near-random chance on coherence-sensitive tasks, and addresses a critical computational bottleneck by offering a 7x training speedup and a 50% reduction in parameters compared to existing coherence models.

Experiments demonstrate a relative accuracy increase of up to 70% over the state-of-the-art and ablations validate the necessity of each component.

Weakness：

While the introduction of phase-equivariance into audio representations is insightful, the model's novelty primarily stems from a strategic application of the fourier shift theorem within a contrastive learning framework, which is not a fundamental breakthrough in general-purpose representation learning architectures.

The current experimental evaluation is heavily concentrated on the task of stem-to-mix retrieval. While the model shows promising results in zero-shot beat tracking and chord probing, a more comprehensive validation of its performance as a general-purpose music representation would require broader cross-comparisons with foundation llm models across a wider variety of downstream music information retrieval tasks.

In the zero-shot beat tracking experiments, the model relies on a synthetic metronome probe to perform similarity calculations. This methodology is essentially a form of template matching within the latent space, and its robustness in handling complex time signatures or music with significant rhythmic fluctuations(such as non-isochronous) has not been fully demonstrated.

typos: 'stat-of-the-art' in line 061

---

> ### Author Rebuttal · Authors · 2026-03-30
>
> We thank the reviewer for their positive assessment and address their questions below:
>
> ### Non-Isochronous Rhythms, Q1
>
> > *In the zero-shot beat tracking system […] How does this mechanism perform on music with significant timing fluctuations, such as non-isochronous rhythms?*
> >
>
> We appreciate this interesting question. Indeed, PHALAR correctly recovers tempo under non-isochronous meters (e.g., 7/4), but its performance degrades under non-periodic tempo changes.
>
> We tested PHALAR on "Money" by Pink Floyd (126 BPM with a 7/4 signature) and produced two results similar to Fig. 5 of the submission, which can be viewed here: [https://anonymous.4open.science/r/icml_rebuttals_images_phalar-006E](https://anonymous.4open.science/r/icml_rebuttals_images_phalar-006E) (`rfig3-4`). PHALAR recovers the underlying pulse at the correct BPM and gracefully handles the non-isochronous feel (`rfig3`). However, as expected, it is less reliable when the beat itself accelerates or decelerates non-periodically. This can be clearly noted around the 172-second mark when the guitar solo starts and a 4/4 signature is adopted (`rfig4`), introducing a distinct change of pace. We will include this analysis as an appendix and discussion in the final revision of the paper.
>
> ### Benchmarking against time-sensitive architectures, Q2
>
> > *Since CLAP is fundamentally designed for semantic invariance and is expected to fail at structural tasks, have the authors considered benchmarking against other time-sensitive or phase-aware architectures used in source separation?*
> >
>
> We would like to clarify that the introduction of CLAP and CDPAM is meant to demonstrate the fundamental orthogonality between semantic similarity and structural coherence, not to claim superiority over these models. COCOLA remains the primary fair comparison for coherence, and our relative improvement over it is our core claim.
>
> That said, we ran additional experiments using frozen MERT embeddings (`m-a-p/MERT-v1-95M` on Hugging Face) under three configurations trained under the same regime described in Section 4.1:
>
> 1. Global average pooling + cosine similarity (`MERT-freeze`).
> 2. Global average pooling + trainable MLP head + bilinear similarity (`MERT-avg`).
> 3. Learned spectral pooling + trainable CVNN head + complex bilinear similarity (`MERT-cplx`).
>
>
>     | Dataset | PHALAR | `MERT-freeze` | `MERT-avg`  | `MERT-cplx` |
>     | --- | --- | --- | --- | --- |
>     | MoisesDB k=8 | 86.79 | 14.06 | 63.53 | 67.39 |
>     | MoisesDB k=16 | 81.49 | 7.63 | 50.58 | 59.13 |
>     | MoisesDB k=64 | 70.87 | 1.83 | 27.82 | 45.85 |
>     | CCS k=8 | 99.65 | 6.44 | 92.36 | 96.49 |
>     | CCS k=16 | 99.45 | 2.37 | 86.41 | 93.79 |
>     | CCS k=64 | 98.61 | 0.31 | 68.74 | 86.65 |
>     | Slakh k=8 | 87.69 | 15.77 | 63.81 | 66.70 |
>     | Slakh k=16 | 83.28 | 8.99 | 52.41 | 58.39 |
>     | Slakh k=64 | 72.37 | 3.35 | 32.64 | 46.13 |
>
> As expected, `MERT-freeze` (which represents the MERT variant of our CLAP and CDPAM tests) fails to solve the task. On the other hand, adding a trainable head drastically improves performance, especially when using our spectral aggregation technique. Still, `MERT-cplx` reaches an accuracy of $45.85\%$ on MoisesDB K=64; which is higher than COCOLA ($41.84\%$), but $24$ points lower than PHALAR.
> This indicates that, to achieve the same results as PHALAR, a MERT model equipped with learned spectral pooling would require full retraining. Because MERT is $40\times$larger than PHALAR, this would require a significant investment of time and computational resources. We remain available for any further inquiries regarding cross-architecture comparisons.
>
> Finally, we thank the reviewer for pointing out the typo on line 061; it will be fixed.

---

> > ### Author Rebuttal · Reviewer_zxvh · 2026-04-03
> >
> > Thanks, I will maintain my score.

---

> > > ### Author Response · Authors · 2026-04-03
> > >
> > > Thank you for your constructive feedback throughout the review process and for confirming that our rebuttal addressed your concerns. The MERT cross-comparisons and the model's behavior under non-isochronous meters gave us a nice opportunity to further test the boundaries of PHALAR. We will make sure these new experiments are integrated into the final manuscript. Many thanks for the time and thought you put into reviewing our submission!

---

### Decision · Program_Chairs · 2026-04-30

**Decision:**

Accept (regular)

**Comment:**

PHALAR proposes a phase-aware contrastive learning model for music stem retrieval, claiming state-of-the-art results through phasor-based representations. Scores range from 4–6 (avg. 4.5), with strong consensus for acceptance. The rebuttal was effective, leading multiple reviewers to raise their scores.

Core strengths were consistently recognized: the motivation for incorporating phase information is well-argued (r7tm, sxzi), the empirical results and ablation studies are strong (u65s, zxvh), and the paper addresses a meaningful gap in existing foundation models like CLAP, which the work demonstrates are blind to phase coherence (zxvh). Computational efficiency relative to existing coherence models was also highlighted (zxvh).

The main substantive weaknesses cluster around scope and comparison fairness. Zxvh notes that phase-aware audio representations are not novel in themselves, and the model is evaluated only on stem retrieval, leaving its value as a general-purpose music representation undemonstrated. Sxzi raises a pointed architectural question — whether the model should be pitch-invariant or pitch-equivariant for harmonic coherence tasks — which was clarified in the rebuttal. Both zxvh and sxzi flagged that baseline comparisons were somewhat unfair, pitting PHALAR against models designed for similarity rather than coherence; the authors addressed this by benchmarking against time-sensitive architectures and frozen MERT embeddings, showing consistent outperformance. The zero-shot beat tracking experiments rely on a synthetic metronome probe and may not generalize to non-periodic tempo changes, a limitation the authors acknowledged. U65s noted the method section lacks detail and the introduction undersells the research problem, concerns addressed in the rebuttal along with disclosure of key failure cases (e.g. compressed audio).

Recommendation: Accept. The contribution is technically sound and the rebuttal substantially addressed the panel's concerns. The final version should expand the method section, include a substantive limitations and future work discussion (sxzi's concern remains valid), and the authors should consider at least one evaluation beyond stem retrieval to support the broader representation learning framing.